# Transcriptomic and Metabolomic Analysis of Korean Pine Cell Lines with Different Somatic Embryogenic Potential

**DOI:** 10.3390/ijms232113301

**Published:** 2022-11-01

**Authors:** Chunxue Peng, Fang Gao, Iraida Nikolaevna Tretyakova, Alexander Mikhaylovich Nosov, Hailong Shen, Ling Yang

**Affiliations:** 1State Key Laboratory of Tree Genetics and Breeding, School of Forestry, Northeast Forestry University, Harbin 150040, China; 2State Forestry and Grassland Administration Engineering Technology Research Center of Korean Pine, Harbin 150040, China; 3Laboratory of Forest Genetics and Breeding, V.N. Sukachev Institute of Forest, Siberian Branch of RAS, Krasnoyarsk 660036, Russia; 4Department of Cell Biology, Institute of Plant Physiology K.A. Timiryazev, Russian Academy of Sciences, Moscow 127276, Russia; 5Department of Plant Physiology, Biological Faculty, Lomonosov Moscow State University, Moscow 119991, Russia

**Keywords:** Korean pine, somatic embryogenic potential, transcriptome, metabolome

## Abstract

The embryogenesis capacity of conifer callus is not only highly genotype-dependent, but also gradually lost after long-term proliferation. These problems have seriously limited the commercialization of conifer somatic embryogenesis (SE) technology. In this study, the responsive SE cell line (R-EC), the blocked SE cell line (B-EC), and the loss of SE cell line (L-EC) were studied. The morphological, physiological, transcriptomic, and metabolomic profiles of these three types of cells were analyzed. We found that R-EC had higher water content, total sugar content, and putrescine (Put) content, as well as lower superoxide dismutase (SOD) activity and H_2_O_2_ content compared to B-EC and L-EC. A total of 2566, 13,768, and 13,900 differentially expressed genes (DEGs) and 219, 253, and 341 differentially expressed metabolites (DEMs) were found in the comparisons of R-EC versus B-EC, R-EC versus B-EC, and B-EC versus L-EC, respectively. These DEGs and DEMs were mainly found to be involved in plant signal transduction, starch and sugar metabolism, phenylpropane metabolism, and flavonoid metabolism. We found that the AUX1 and AUX/IAA families of genes were significantly up-regulated after the long-term proliferation of callus, resulting in higher auxin content. Most phenylpropane and flavonoid metabolites, which act as antioxidants to protect cells from damage, were found to be significantly up-regulated in R-EC.

## 1. Introduction

Somatic embryogenesis (SE) is currently one of the most promising methods for the reproduction of commercially viable conifers, since it can rapidly produce large quantities of high-value seedlings with desirable characteristics [1,2]. Korean pine is an economically and ecologically important conifer and mainly distributed in Northeastern China. Embryogenic callus was successfully obtained by using immature embryos of Korean pine, with an ultra-low temperature preservation and somatic embryo maturation system successfully established [3]. However, there are still many problems that hinder the large-scale application of this technique. For example, there is a strong genotype dependence in the process of the SE of Korean pine, and the SE ability of callus is difficult to maintain for long periods of time [4,5]. Similar issues have been noted in other species. In *Gossypium*, higher levels of fatty acid, tryptophan, and pyruvate metabolism were found in cell lines with SE ability, and transcription factors related to SE were significantly preferentially activated during somatic embryo differentiation [6]. In *Araucaria angustifolia*, a high accumulation of putrescine (Put) and a small amount of ethylene were found in cell lines with low SE potential, whereas higher ethylene and reactive oxygen species (ROSs) were observed in cell lines with high SE potential [7]. To date, no studies have elucidated the reasons for the differences in the SE potential of different genotypes under the same in vitro culture conditions. In order to comprehensively study the molecular mechanism of SE, it is crucial to explore the dynamic transcriptional maps and gene regulation patterns involved. With long-term proliferation of embryogenic callus, SE ability gradually declines before eventually disappearing completely [8]. In coastal pine and larch, embryogenic callus loses SE capacity after six months of proliferation [9,10]. In *Eucalyptus* and hybrid larch (*Larix eurolepis* and *Larix marschlinsii*), embryogenic callus can still produce somatic embryos after three and nine years of proliferation, respectively [11,12].

It is essential to better understand the physiological and molecular events related to somatic embryo productivity. We selected the responsive SE cell line (R-EC), the blocked SE cell line (B-EC), and the loss of SE cell line (L-EC) for this study. The morphological and physiological differences between these lines were explored, and transcriptomic and metabolomic studies were carried out to identify the key factors that determine the expression of totipotency in somatic cells and lay the foundation for revealing the molecular genetic network underlying totipotency in plant cells.

## 2. Results

### 2.1. Morphological and Physiological Responses between Different Cell Lines

After seven days of proliferation, there was no significant difference in morphology between the three cell lines, which were all white and transparent (Figure 1A). However, significant differences were found when comparing R-EC, L-EC, and B-EC fresh weight, dry weight, water content, starch, total sugar, catalase (CAT), superoxide dismutase (SOD), Put, spermine (Spm), spermidine (Spd), indole-3-acetic acid (IAA), abscisic acid (ABA), and H_2_O_2_ content. R-EC had the highest fresh weight, water, total sugar, and Put content, while B-EC had the highest levels of starch, CAT, SOD, and IAA, and L-EC had the highest dry weight, Spm, Spd, IAA, and H_2_O_2_ content (Figure 1B).

### 2.2. Transcriptome Analysis among R-EC, L-EC, and B-EC

The transcriptomes of R-EC, B-EC, and L-EC were analyzed to investigate the molecular events between different embryogenic calli. After removing low-quality reads, a total of 546,004,658 clean reads were obtained. The detection rates of Q20 and GC were 97.06–97.19% and 44.16–44.27%, respectively, indicating a higher quality of transcriptome sequencing data (Table 1).

A total of 264,676 Unigenes were assembled from clean sequences using Trinity, of which 1842 were N50 non-redundant sequences and 459 were N90 non-redundant sequences (Table 2); the 300–400 bp length accounted for the largest proportion, and simultaneously, the assembled transcript and Unigene decreased gradually with increasing sequence length (Figure 2).

#### 2.2.1. Analysis of Differentially Expressed Genes DEGs in R endash EC, L endash EC, and B endash EC

We created Venn diagrams to identify overlapping genes in three different types of Korean pine callus. There were 321 DEGs in common between R-EC versus (vs.) B-EC, R-EC vs. L-EC, and B-EC vs. L-EC (Figure 3A). Additionally, the tissues of the three cell lines of Korean pine with different SE potential were compared to identify the DEGs. There were 2566 DEGs in total between R-EC and L-EC, of which 858 DEGs were up-regulated and 1708 DEGs were down-regulated. There were 13,768 DEGs in total between R-EC and B-EC, of which 6951 DEGs were up-regulated and 6817 DEGs were down-regulated. Finally, there were 13,900 DEGs between B-EC and L-EC, of which 6516 DEGs were up-regulated and 7384 DEGs were down-regulated (Figure 3B). To further evaluate the biological functions of DEGs in cell lines with different SE potential, KEGG enrichment analysis was performed on the DEGs. The DEGs in the R-EC vs. L-EC comparison were annotated into 122 KEGG metabolic pathways, among which 14 metabolic pathways were significantly enriched, including phenylpropanoid biosynthesis, amino sugar and nucleotide sugar metabolism, flavonoid biosynthesis, circadian rhythm, biosynthesis of secondary metabolites, as well as starch and sucrose metabolism. The DEGs in the R-EC vs. B-EC comparison were annotated into 137 KEGG metabolic pathways, of which 16 were significantly enriched, including glycosylphosphatidylinositol (GPI)-anchored biosynthesis, phenylpropanoid biosynthesis, starch and sucrose metabolism, and other glycan degradation (Figure 3B). DEGs in the B-EC vs. L-EC comparison were annotated into 137 KEGG metabolic pathways, of which 22 were significantly enriched, including GPI-anchored biosynthesis, metabolic pathways, indole alkaloid biosynthesis, pentose and glucuronate interconversions, and pantothenate and CoA biosynthesis (Figure 3C).

#### 2.2.2. Transcription Factor Expression among R-EC, L-EC, and B-EC

A total of 135 transcription factors (TFs) (39 up-regulated and 96 down-regulated) were identified in the R-EC vs. L-EC comparison. A total of 660 TFs (272 up-regulated and 388 down-regulated) were identified in the R-EC vs. B-EC comparison. A total of 634 TFs (323 up-regulated and 311 down-regulated) were identified in the B-EC vs. L-EC comparison (Figure 4).

According to several reports [13,14,15], the WRKY, MYB, and AP2-ERF families of genes play key roles in the process of SE. Therefore, the expression levels of DEGs in the WRKY, MYB, and AP2-ERF families were analyzed. In the R-EC vs. L-EC comparison, there were 4 WRKY family genes (all down-regulated), 14 MYB family genes (1 up-regulated and 13 down-regulated), and 16 AP2/ERF-ERF family genes (2 up-regulated and 14 down-regulated). In the R-EC vs. B-EC comparison, there were 19 WRKY family genes (4 up-regulated and 15 down-regulated), 21 MYB family genes (6 up-regulated and 15 down-regulated), and 53 AP2/ERF-ERF family genes (27 up-regulated and 26 down-regulated). In the B-EC vs. L-EC comparison, there were 19 WRKY family genes (6 up-regulated and 13 down-regulated), 30 MYB family genes (13 up-regulated and 27 down-regulated), and 46 AP2/ERF-ERF family genes (20 up-regulated and 26 down-regulated) (Table 3).

Based on these results, we analyzed the expression levels of all WRKY family genes across R-EC, L-EC, and B-EC. A total of 14 WRKY family genes were most highly expressed in R-EC; three were most highly expressed in L-EC; 16 were most highly expressed in L-EC (Figure 5a). Among the WYB family of genes, 20 were most highly expressed in R-EC, three were most highly expressed in L-EC, and 17 were most highly expressed in B-EC (Figure 5b). Among the AP2-ERF-ERF family of genes, 25 were most highly expressed in R-EC, 16 were most highly expressed in L-EC, and 33 were most highly expressed in B-EC (Figure 5c,d).

### 2.3. Metabolomic Analysis of R-EC, L-EC, and B-EC

The metabolomes of the three cell lines were profiled using an LC-ESI-MS/MS system. A total of 1013 metabolites were identified in all samples, including 875 overlapping metabolites, four unique metabolites in R-EC, six in L-EC, and 56 in B-EC (Figure 6A). These metabolites were classified into 15 types, of which flavonoids (23.9%), lipids (11.9%), and amino acids and their derivatives (11.3%) accounted for the highest proportions (Figure 6B). We next analyzed the distribution of metabolites in the cell lines and found that flavonoids, phenolic acids, nucleotides and their derivatives, terpenes, stilbene, and tannins were highly accumulated in B-EC. Quinones were highly accumulated in R-EC, while organic acids were highly accumulated in L-EC. In addition, flavonoids, phenolic acids, stilbene, and tannins were slightly higher in L-EC (Figure 6C).

#### 2.3.1. Differential Metabolite Analysis in R-EC, L-EC, and B-EC

A total of 458 differentially expressed metabolites (DEMs) were identified in R-EC, L-EC, and B-EC. Among them, there were 39 DEMs in R-EC, L-EC, and B-EC, with 47 DEMs unique to R-EC and B-EC, 35 DEMs unique to B-EC and L-EC, and 62 DEMs unique to B-EC and L-EC (Figure 7A). Additionally, 219 DEMs (25 up-regulated and 194 down-regulated) were identified in the R-EC vs. L-EC comparison, and 252 DEMs (183 up-regulated and 7 down-regulated) were identified in the R-EC vs. B-EC comparison. A total of 340 DEMs (41 up-regulated and 300 down-regulated) were identified in the B-EC vs. L-EC comparison (Figure 7B).

Next, KEGG enrichment analysis was performed on the DEMs. The DEMs in the R-EC vs. L-EC comparison were assigned to 45 pathways, with significant enrichment found in flavonoid biosynthesis, phenylpropane biosynthesis, and stilbenoid, diarylheptanoid, and gingerol biosynthesis (Figure 7A). The DEMs in the R-EC vs. B-EC comparison were assigned to 61 pathways, with significant enrichment in isoflavone biosynthesis, alpha-linolenic acid metabolism, and secondary metabolite biosynthesis (Figure 7D). DEMs in the B-EC vs. L-EC comparison were assigned to 70 pathways, with significant enrichment found in phenylpropane biosynthesis, flavonoid biosynthesis, and arginine and proline metabolism (Figure 7E).

#### 2.3.2. Functional Analysis of DEGs and DEMs among R-EC, L-EC, and B-EC

##### Differences in Hormone Signaling Function between Different Cell Lines

Through metabolomics analysis, we found that auxin content was highest in L-EC and lowest in B-EC. Zeatin (ZT) and gibberellinc acid (GA) were not significantly different among R-EC, B-EC, and L-EC, while ABA, salicylic acid (SA), and jasmonic acid (JA) were highly abundant in B-EC. At the transcriptional level, we identified 219 DEGs involved in phytohormone signaling, with 62, 27, 54, 18, 44, and 14 DEGs in the IAA, CTR, GA, ABA, JA, and SA signaling pathways, respectively. In the IAA signaling pathway, most AUX1 and AUX/IAA genes were highly expressed in L-EC, while TIR1 genes were highly expressed in R-EC, and SAUR genes were highly expressed in R-EC. In the ZT synthesis pathway, most of the CRE1 and A-ARR genes were highly expressed in R-EC and L-EC. In the GA synthesis pathway, most GID1 genes were highly expressed in B-EC, most DELLA genes were highly expressed in R-EC and B-EC, and most TF genes were highly expressed in L-EC and B-EC. In the ABA synthesis pathway, seven PYR-/PYL-related genes were highly expressed in R-EC, two were highly expressed in B-EC, and no gene was highly expressed in B-EC. In the SA synthesis pathway, JAR1 and MYC2-related genes were highly expressed in R-EC, and JAZ genes were highly expressed in B-EC (Figure 8).

##### Differences in Starch and Sugar Metabolism between Different Cell Lines

At the metabolic level, sucrose and UDP-glucose were found to be highly accumulated in B-EC, while D-fructose, D-glucose, and D-maltose were found to be highly accumulated in L-EC. Glucose-1-phosphate content was higher in R-EC, L-EC, and B-EC, while D-glucose-6-p and D-fructose-6-p were highly accumulated in R-EC and L-EC. Most of the granule-bound starch synthase genes were highly expressed in B-EC at the transcriptional level, and most 1,4-α-glucan branching enzyme genes were highly expressed in L-EC. Most α-amylase genes and β-fructofuranosidase genes were highly expressed in R-EC and B-EC, while most β-amylase genes were highly expressed in L-EC (Figure 9).

##### Differences in Phenylpropane Metabolic Pathways between Different Cell Lines

At the metabolic level, R-EC, B-EC, and L-EC showed significant differences in metabolic pathways associated with phenylpropane. Among them, the contents of phenylalanine, caffeoyl shikimic acid, p-coumaric acid, caffeic acid, p-coumaraldehyde, and coniferin were highest in B-EC. P-coumary alcohol, scopoline, and coniferyl alcohol were most highly expressed in R-EC, while in L-EC, all metabolites were expressed at low levels. At the transcriptional level, most of the phenylalanine ammonia lyase genes, cinnamoyl-CoA reductase genes, and cinnamyl alcohol dehydrogenase genes were highly expressed in B-EC. Most of the *trans*-cinnamic acid 4-monooxygenase and coniferyl alcohol glucosyltransferase genes were highly expressed in R-EC (Figure 10).

##### Differences in Flavonoid Metabolic Pathways among Different Cell Lines

At the metabolic level, phloretin, hesperetin-7-O-glucoside, naringenin chalcone, vitexin, apigenin, and luteolin were highly expressed in B-EC, while prunin, naringenin, dihydrokaempferol, afzelechin, and epicatechin were highly expressed in R-EC. Phloridzin was highly expressed in L-EC. At the transcriptional level, chalcone synthase, flavanone 4-reductase, pomelo peel 3-dioxygenase, flavonoid 3′,5′-hydroxylase, and anthocyanin reductase genes were expressed at high levels in R-EC. Chalcone isomerase and flavonoid 3′-monooxygenase genes were highly expressed in B-EC, while the flavanone 7-O-glucoside 2″-O-β-L-rhamnosyltransferase gene was highly expressed in L-EC (Figure 11).

## 3. Discussion

We conducted transcriptomic analysis of three Korean pine cell lines and obtained 54,600,4658 clean reads. In the R-EC vs. L-EC comparison, 2566 DEGs were identified and assigned to 122 KEGG pathways. In the R-EC vs. B-EC comparison, 13,768 DEGs were identified and assigned to 137 KEGG pathways, and in the B-EC vs. L-EC comparison, 13,900 DEGs were found and assigned to 137 KEGG pathways. The most highly represented pathways were phenylpropane biosynthesis, plant hormone signal transduction, secondary metabolite biosynthesis, flavonoid biosynthesis, and starch and sucrose metabolism. Some studies have shown that phenylpropane biosynthesis plays a critical role in plant embryonic development, and this pathway is known to be related to stress tolerance [16]. Genes related to phenylpropanoid biosynthesis were found to be significantly enriched in embryogenic callus in studies of both *Carica papaya* and *Fragaria* × *ananassa* [17,18]. Plant-hormone-related genes also play a key role in the SE process [19]. Additionally, during the early somatic embryo development of *Dimocarpus longan*, plant hormone-related genes are enriched, especially cytokinin and auxin signaling components [20]. Signal transduction pathways that control sucrose and starch accumulation are critical for somatic embryonic development. The nature of carbohydrate supply can reflect the signaling network that controls development, and endogenous carbohydrate status varies during SE in conifers, which can be used to identify cell lines with high-quality somatic embryos [21]. To better understand differences in metabolite levels among Korean pine cell lines, we investigated the distribution of metabolites in different embryogenic callus (R-EC, B-EC, and L-EC) of Korean pine and identified 1013 metabolites across all tissues. These metabolites were divided into 15 types, of which flavonoids (23.9%), lipids (11.9%), and amino acids and their derivatives (11.3%) accounted for the highest proportions. We found that flavonoids, phenolic acids, nucleotides and their derivatives, terpenoids, stilbene, and tannins were highly accumulated in B-EC, while quinones were highly accumulated in R-EC, and organic acids were highly accumulated in L-EC. In addition, flavonoids, phenolic acid, stilbene, and tannins were only slightly accumulated in B-EC. These data suggest that the accumulation of Korean pine metabolites occurs in a tissue-specific manner, a phenomenon that has been observed in other plant species [22].

### 3.1. Influence of Transcription Factors during Somatic Embryogenesis

TFs have been shown to play key roles in plant embryogenesis and development in numerous species. Studies of somatic embryo development have shown that a complex network of transcriptional regulation maintains embryogenic capacity and guides embryogenic callus formation [23]. According to several reports, the WRKY, MYB, and AP2-ERF families of genes play key roles in the process of SE, and we found that most of the DEGs in the WRKY, MYB, and AP2-ERF families were highly expressed in the R-EC and B-EC cell lines. Studies have also shown that the WRKY family of genes is up-regulated during the formation of embryogenic callus in *wheat* [24], papaya, and *Arabidopsis* [25]. The MYB family is also involved in plant development, growth [26], and hormone signal transduction [27], while the AP2-EREBP TF family plays an important role in cell proliferation and embryogenesis [28]. The up-regulated expression of the MYB and AP2-EREBP families of genes in R-EC and B-EC cell lines in Korean pine indicates that young embryogenic callus has higher viability than old embryogenic callus. In addition, L-EC showed significant up-regulation of three, one, and ten DEGs in the WRKY, MYB, and AP2-ERF families, respectively. This finding indicated that these genes may be related to embryogenic callus senescence and loss of SE.

### 3.2. Effects of Plant Hormones on Somatic Embryogenesis Potential

Plant hormone signal transduction plays a crucial role during SE [29]. According to our findings, many DEGs related to hormone metabolism (auxin, cytokinin, GA, ABA, ethylene, brassinolide, and SA) and signaling pathways were differentially expressed when comparing R-EC, B-EC, and L-EC. We also found that IAA content was highest in L-EC and lowest in B-EC. A similar phenomenon was reported in a study of *L. sibirica*, where the IAA content of embryogenic callus was found to gradually increase with long-term proliferation [30]. Some studies have noted that the content of endogenous auxin increases with the increase of exogenous auxin [31]. Therefore, the accumulation of IAA content after long-term proliferation of embryogenic callus may be due to the accumulation of a large amount of auxin. We found that most AUX1 and AUX/IAA genes were up-regulated in L-EC, and the AUX/IAA and SAUR genes have been identified as early auxin-responsive genes, which can precisely and quickly regulate downstream genes to alter plant growth and development [32]. ABA plays an important role in the accumulation of nutrients during somatic embryo development and maturation [33]. The ABA content in B-EC was significantly higher than that in R-EC and L-EC. Similar findings have been reported in *Hevea brasiliensis* [34] and alfalfa [35], where embryogenic callus showed lower ABA levels than non-embryogenic callus. At the transcriptional level, seven PYR/PYL-related DEGs were highly expressed in R-EC, while two were highly expressed in B-EC, and no associated genes were highly expressed in B-EC. JA and SA also play key roles in the process of SE. At the metabolic level, JA and SA contents in B-EC were significantly higher compared to those in R-EC and L-EC, and there was no significant difference between R-EC and L-EC. At the transcriptional level, JAZ was highly expressed in B-EC, while MYC2, NPR1, and PR-1 were highly expressed in R-EC. Overall, these results indicate that the phytohormone signaling pathways may be key regulators of SE.

### 3.3. The Role of Starch and Sugar Metabolism in Somatic Embryogenesis

The maltase-glucoamylase, nucleotide diphosphatase, starch synthase, and β-amylase genes were significantly up-regulated in L-EC, while the granule-bound starch synthase and hexokinase genes were significantly up-regulated in B-EC. These findings indicated that starch and sugar metabolism were significantly different among R-EC, B-EC, and L-EC and that genes related to starch and sugar metabolism were mainly up-regulated in B-EC and L-EC. In *Tulipa gesneriana*, it has been shown that α-amylase (AMY) and β-amylase (BMY) are the main enzymes involved in the starch degradation process, and an increase in amylase activity results in accelerated starch degradation [36]. Our results indicated that the amylase activity in B-EC and L-EC was higher than that in R-EC, further reinforcing the idea that starch content is important in SE. In addition, we found that sugars and alcohols were highly accumulated in B-EC compared to L-EC. Carbohydrate levels were also found to be higher in non-embryonic callus compared to embryogenic callus in mangosteen studies, with fructose, glucose, and sucrose accumulating most significantly in non-embryonic structures from seed and leaf cultures [37].

### 3.4. The Role of Phenylpropane Metabolism in Somatic Embryogenesis

The phenylpropanoid biosynthetic pathway plays a critical role during both abiotic and biotic stress and is thought to produce many antioxidants, including flavonoids, phenols, lignin, and lignin precursors, which affect plant–pathogen interactions [38,39]. We found that R-EC, B-EC, and L-EC differed significantly in metabolic pathways involving phenylpropane. Among them, phenylalanine, caffeoylshikimic acid, p-coumaric acid, caffeic acid, p-coumaraldehyde, and conifers were highly expressed in B-EC, p-coumaryl alcohol, scopoline, and coniferyl alcohol were highly expressed in R-EC, and in L-EC, all metabolites were present at low levels. Scopolamine and caffeic acid are the precursors of lignin, which is a phenylpropanoid compound found in secondary cell walls. Lignin is the second-most abundant biopolymer in plants, playing roles in plant growth and development, cellular mechanical support, and responses to biotic or abiotic stresses [40,41,42]. This finding suggests that R-EC and B-EC may have a stronger differentiation capacity than L-EC.

### 3.5. The Role of Flavonoid Metabolism in Somatic Embryogenesis

Flavonoids are the third-largest group of natural products and the most diverse secondary metabolites derived from polyphenols [43]. They are involved in plant–environment interactions and defense responses against pathogens, UV radiation, abiotic stresses, and more [44]. In our study, significant differences were found in flavonoid metabolic pathways at both the transcriptomic and metabolic levels. Similar results have been found in citrus, where no detectable accumulation of flavonoids was found in undifferentiated callus, but flavonoids were accumulated after embryonic morphological changes [45]. Other studies showed that flavonoids produced during SE in milk thistle may stimulate differentiation and create a more favorable environment for embryogenesis [46]. In cotton SE studies, it was found that the biosynthesis of flavonoids was associated with somatic embryo development during the transdifferentiation of somatic embryos [6]. These findings indicate that the flavonoid anabolic pathway plays an important role in SE. In addition, some studies have found that the antioxidant activity of flavonoids may contribute to the induction of embryogenic callus and SE [24]. In our study, most of the differential metabolites found in comparisons among R-EC, B-EC, and L-EC are known to act as strong oxidants, and other studies have found that redox changes in seeds and embryos control plant growth and development [47]. These findings imply that the different SE ability of R-EC, B-EC, and L-EC may be closely related to redox reactions.

## 4. Materials and Methods

### 4.1. Initiation of Korean Pine Embryogenic Callus

The plants used in this study complied with international guidelines. Full sibling family cones 1# were authorized to be collected from cooperative institution (Korean pine seed orchard of Lushuihe Forestry Bureau of Jilin Province) on 1 July 2018. The SE cell line (R-EC), the blocked SE cell line (B-EC), and the loss of SE cell line (L-EC) were used as research materials. They were all induced by 1# family. Embryonic callus was initiated based on the method of Peng et al. [3]. Briefly, synchronized callus was transferred to solid proliferation medium for culture (DCR [48], supplemented with 2.3 μM·L^−1^ 2,4-dichlorophenoxyacetic acid (2,4-D), 0.5 μM·L^−1^ 6-benzyladenine (6-BA), 0.5 g·L^−1^ L-glutamine, 0.5 g·L^−1^ casein hydrolysate, and 0.4 g·L^−1^ gellan gum). After seven days of proliferation, materials were collected for morphological observation, physiological examination, transcriptome sequencing, and metabolome analysis.

### 4.2. Experimental Methods

#### 4.2.1. Morphological Observation

After seven days of embryonic tissue proliferation, the differences between callus samples were observed under a stereo microscope (SZX-ILLB2-200, Tokyo, Japan).

#### 4.2.2. Physiological Measurements

After seven days of embryonic tissue proliferation, materials were collected for the examination of the physiological parameters, in which the contents of soluble sugar, soluble protein, and starch were determined according to the methods described by Peng et al. [5]. The determination of IAA and ABA contents was carried out following the method described by Li and Peng [49,50]. Approximately 0.1 g fresh weight (FW) of the sample was homogenized in liquid nitrogen and extracted in cold 80% (*v*/*v*) methanol with butylated hydroxytoluene (1 mmol·L^−1^) overnight at 4 °C. The supernatant was collected after centrifugation at 1000× *g* (4 °C) for 10 min, passed through a C18 Sep-Pak cartridge, and dried with nitrogen. The residue was dissolved in phosphate-buffered saline (0.01 mol·L^−1^, pH 7.4), and the contents of IAA and ABA were determined. Put, Spd, and Spm were determined according to the method of Neusa Steiner [51]. Briefly, approximately 0.2–0.3 g FW of the sample was extracted in 5% (*w*/*v*) HClO_4_, and 10 mL of the culture filtrate was freeze-dried and dissolved in 5% (*w*/*v*) HClO_4_ to analyze free and soluble conjugated polyamine from the supernatant of the centrifuged extracts. The polyamine dansyl derivatives were analyzed by high-performance liquid chromatography (HPLC). Each measurement was repeated three times.

#### 4.2.3. Transcriptomic Analysis

After seven days of proliferation, the materials were sampled for transcriptome sequencing. Total RNA was extracted from the samples using TRIzol reagent according to the manufacturer’s instructions and then characterized on a 1% agarose gel. RNA was checked for purity using a NanoDrop 2000 spectrophotometer, and an Agilent 2100 Bioanalyzer was used to examine RNA integrity. The RNA extractions of the samples were mixed in equal amounts, followed by library construction. A total of 3 μg of RNA per sample was used as the input material for cDNA library construction. mRNA was enriched and purified with oligonucleotide (dT)-rich magnetic beads, followed by fragmentation. Using these cleaved mRNA fragments as templates, first-strand cDNA was synthesized with oligo-dT primers. Second-strand cDNA was synthesized using random primers. The resulting cDNA was then end-repaired and phosphorylated using T4 DNA polymerase and Klenow DNA polymerase. Afterwards, an “A” base was inserted as an overhang at the 3′ end of the repaired cDNA fragments, and Illumina paired-end Solexa adapters were subsequently ligated to the cDNA fragments to distinguish different sequencing samples. To select the size range of templates for downstream enrichment, the product of the ligation reaction was purified and visualized on 2% agarose gel. Next, PCR amplification was performed to enrich the purified cDNA template. RNA sequencing was performed on an Illumina Hiseq 2000 platform. RNA-seq data were quality controlled using SeqPrep and Sickle, with the default parameters. Clean reads were obtained by removing reads containing adapter sequences, more than 1% ambiguous “N” bases, or bases with quality below Q15. All clean data were used for de novo assembly, differentially expressed gene analysis, and functional annotation using Trinity.

#### 4.2.4. Metabolomics Analysis

Extract analysis and metabolite identification and quantification were performed by MetWare (Wuhan, China), following the standard procedures utilized in previous studies [52]. Metabolite data analysis were performed using Analyst 1.6.3 software. Metabolites with a fold change of ≥1.5 were considered DEMs.

## 5. Conclusions

This study analyzed the transcriptomic and metabolomic differences among R-EC, L-EC, and B-EC. A total of 2566, 13,768, and 13,900 DEGs, as well as 219, 253, and 341 DEMs were found in the R-EC vs. B-EC, R-EC vs. B-EC, and B-EC vs. L-EC comparisons, respectively. These DEGs and DEMs were mainly found to be involved in plant signal transduction, starch and sugar metabolism, phenylpropane metabolism, and flavonoid metabolism. These findings significantly expand the current understanding of the mechanisms regulating SE in Korean pine.

## Figures and Tables

**Figure 1 ijms-23-13301-f001:**
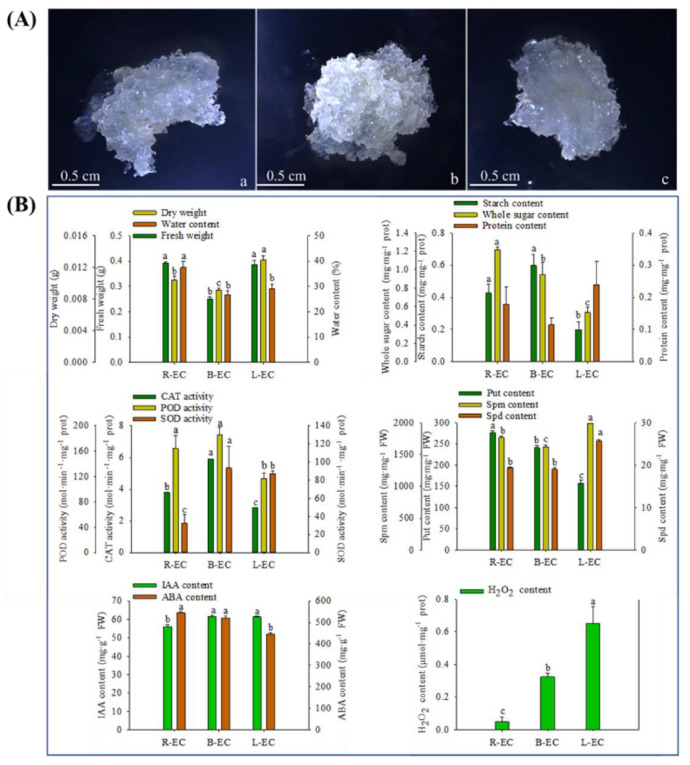
Morphological (**A**) and physiological (**B**) differences among Korean pine R-EC, B-EC, and L-EC. (**a**) R-EC. (**b**) B-EC. (**c**) L-EC. The scale bar represents 0.5 cm.

**Figure 2 ijms-23-13301-f002:**
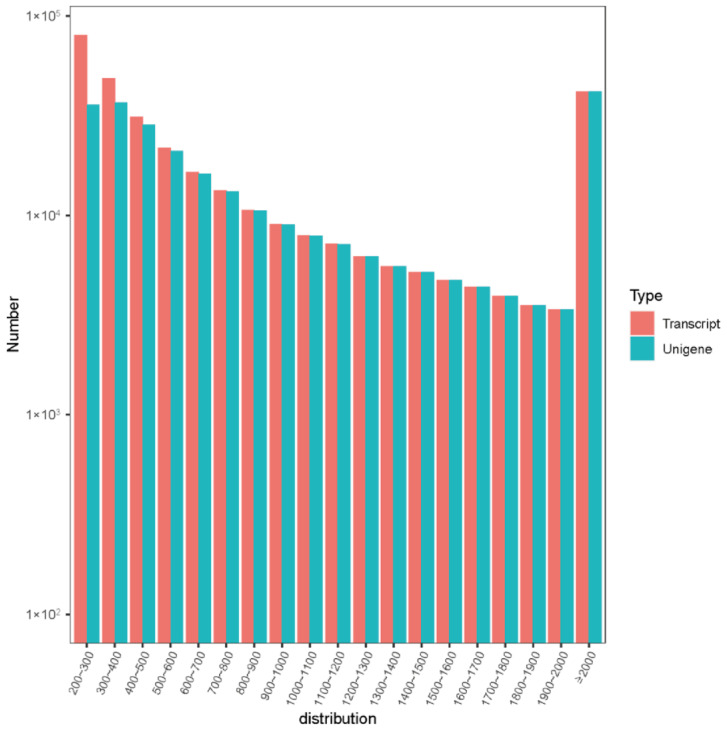
Splicing sequence length distribution map of Korean pine transcripts.

**Figure 3 ijms-23-13301-f003:**
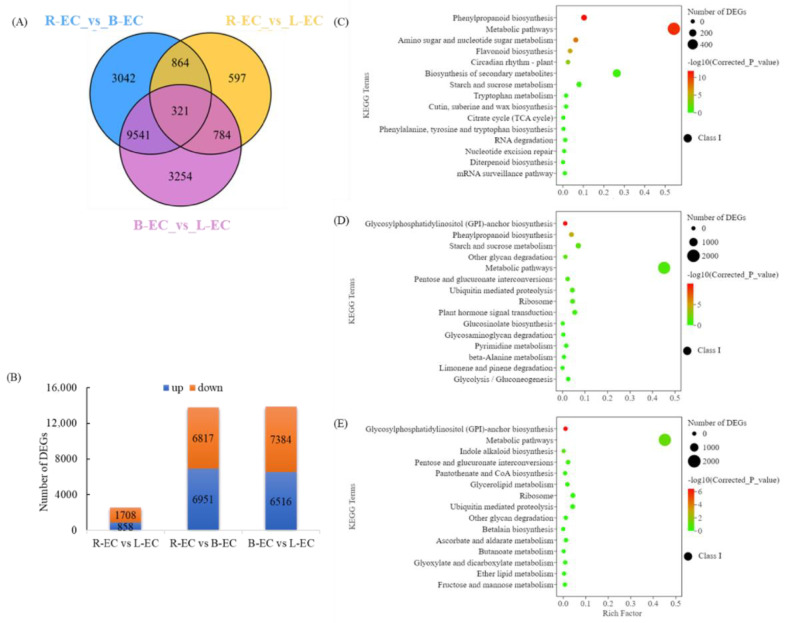
Transcriptomic analysis of Korean pine R-EC, B-EC, and L-EC. (**A**) Venn diagram showing the number of DEGs that overlap between different tissues. (**B**) Statistics on the number of DEGs between different tissues. (**C**–**E**) DEGs enriched in different KEGG pathways.

**Figure 4 ijms-23-13301-f004:**
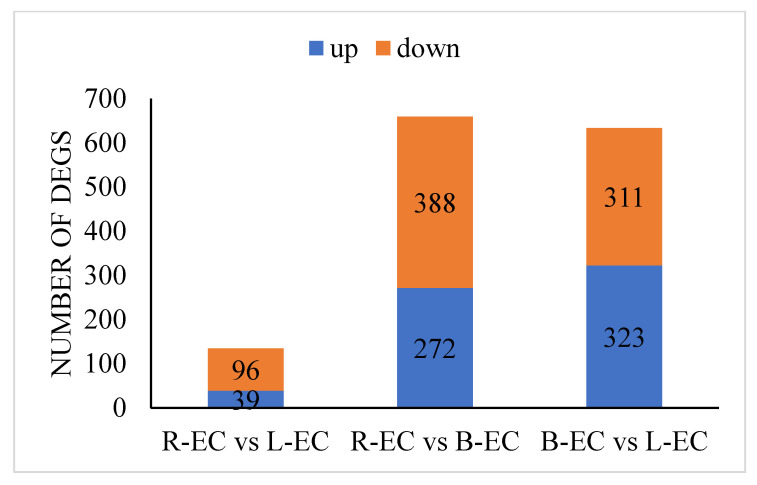
Korean pine transcription factor expression statistics.

**Figure 5 ijms-23-13301-f005:**
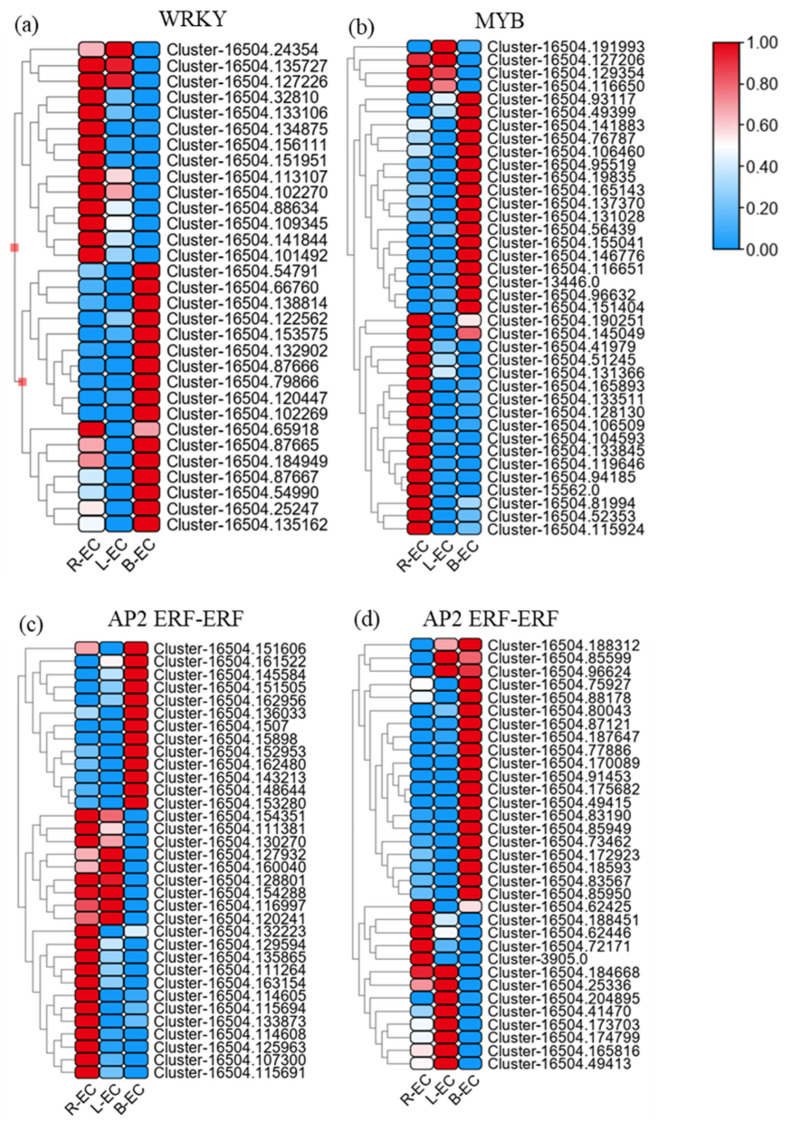
Differences in the expression levels of the WRKY, MYB, and AP2-ERF-ERF families of genes between different Korean pine cell lines. Heatmap indicate the gene expression level by Log2 [FPKM] with a rainbow color scale.

**Figure 6 ijms-23-13301-f006:**
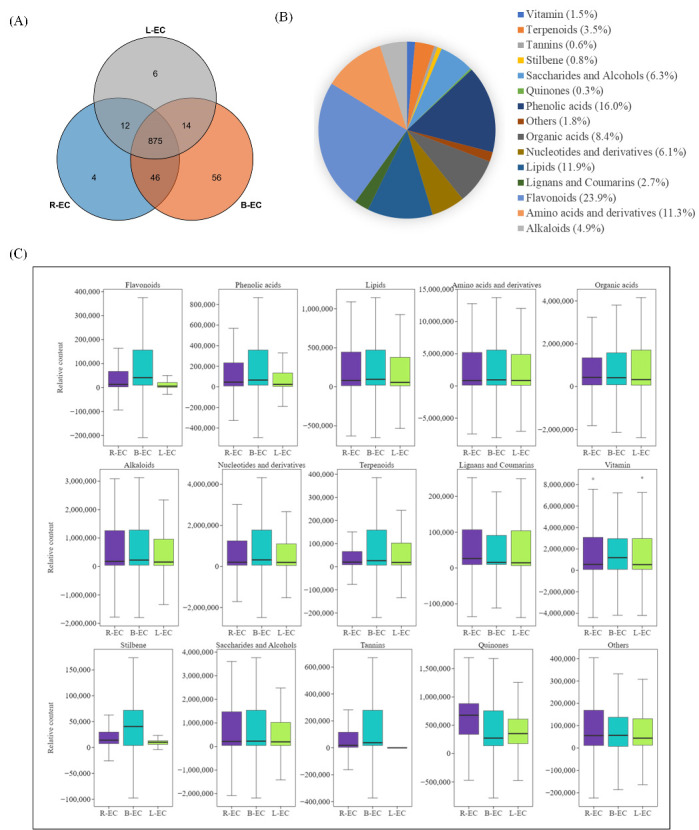
Metabolomic analysis of Korean pine R-EC, B-EC, and L-EC cell lines. (**A**) Quantitative statistics of metabolites among R-EC, B-EC, and L-EC. (**B**) R-EC, B-EC, and L-EC metabolite classification. (**C**) Variability of different metabolite classes in R-EC, B-EC, and L-EC.

**Figure 7 ijms-23-13301-f007:**
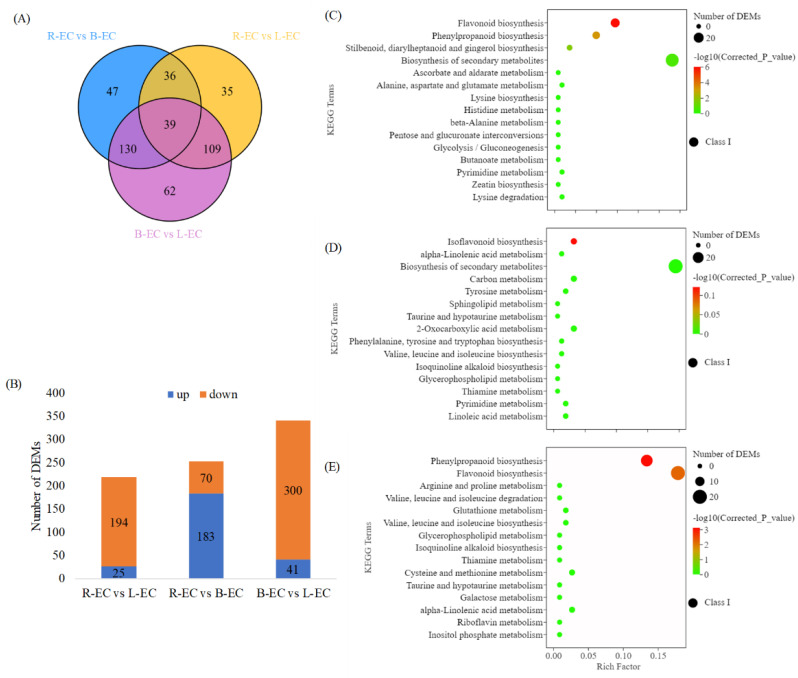
Statistical analysis of differentially expressed metabolites (DEMs) among R-EC, L-EC, and B-EC Korean pine cell lines. (**A**) Venn diagram showing the number of DEMs between different tissues. (**B**) Statistics on the number of DEMs when comparing different tissues. (**C**–**E**) DEMs enriched in different KEGG pathways.

**Figure 8 ijms-23-13301-f008:**
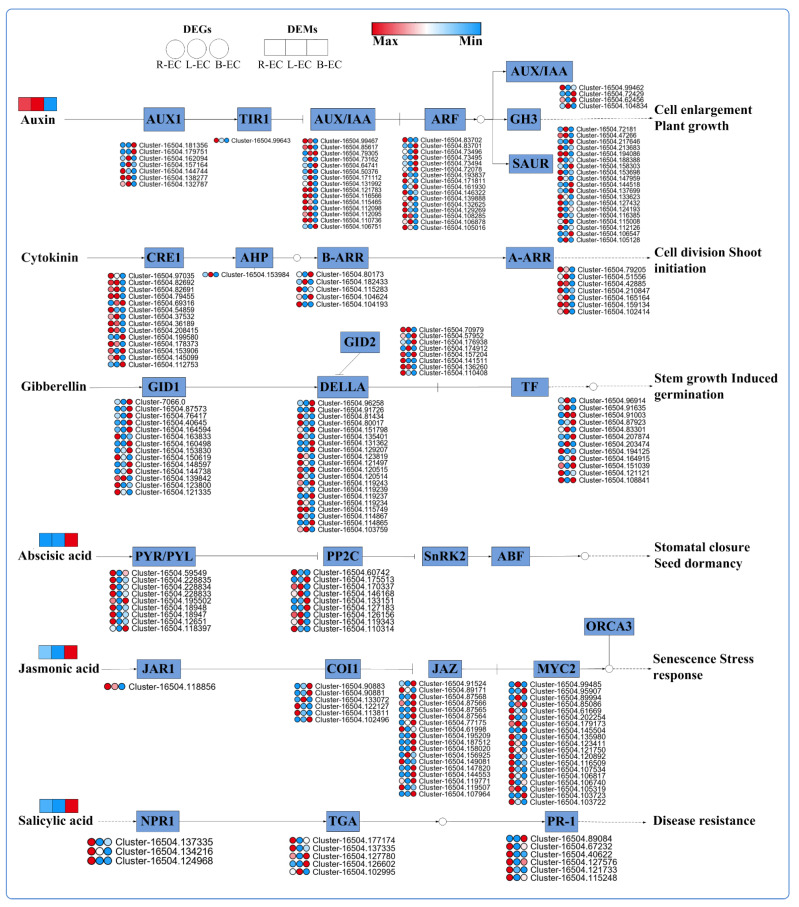
Analysis of plant signal transduction pathways among Korean pine R-EC, L-EC, and B-EC cell lines.

**Figure 9 ijms-23-13301-f009:**
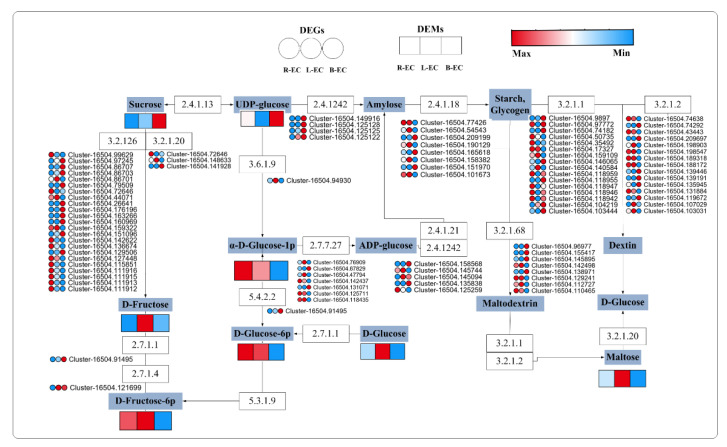
Analysis of DEGs and DEMs from starch and glucose metabolism pathways among Korean pine R-EC, L-EC, and B-EC cell lines. 3.1.1.26: beta-fructofuranosidase, 3.2.1.20: maltase-glucoamylase, 2.4.1.13: sucrose synthase, 3.6.1.9: nucleotide diphosphatase, 2.4.1242: granule-bound starch synthase, 2.4.1.18: 1,4-alpha-glucan branching enzyme, 3.2.1.1: alpha-amylase, 3.2.1.2: beta-amylase. 2.4.1.21: starch synthase, 3.2.1.68: isoamylase, 5.4.2.2: phosphoglucomutase, 2.7.7.27: glucose-1-phosphate adenylyltransferase, 2.7.1.1: hexokinase, 2.7.1.4: fructokinase, 5.3.1.9: glucose-6-phosphate isomerase.

**Figure 10 ijms-23-13301-f010:**
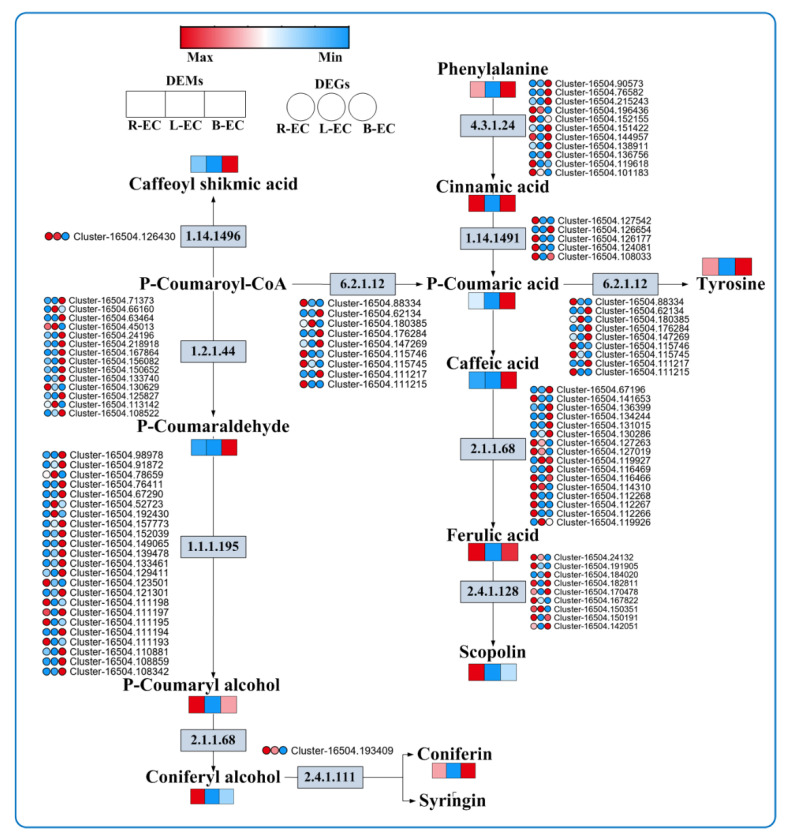
Analysis of DEGs and DEMs of phenylpropane metabolic pathways among Korean pine R-EC, L-EC, and B-EC cell lines. 4.3.1.24: phenylalanine ammonia lyase; 1.14.14.91: trans-cinnamic acid 4-monooxygenase; 6.2.1.12: 4-coumaric acid-CoA ligase; 2.1.1.68: caffeic acid 3-O-methyltransferase; 2.4.1.128: Scopolamine glucosyltransferase; 1.14.14.96: 5-O-(4-coumaroyl)-D-quinic acid 3′-monooxygenase; 1.2. 1.44: cinnamoyl-CoA reductase; 1.1.1.195: cinnamyl alcohol dehydrogenase; 2.1.1.68: caffeic acid 3-O-methyltransferase; 2.4.1.111: coniferyl alcohol glucosyltransferase.

**Figure 11 ijms-23-13301-f011:**
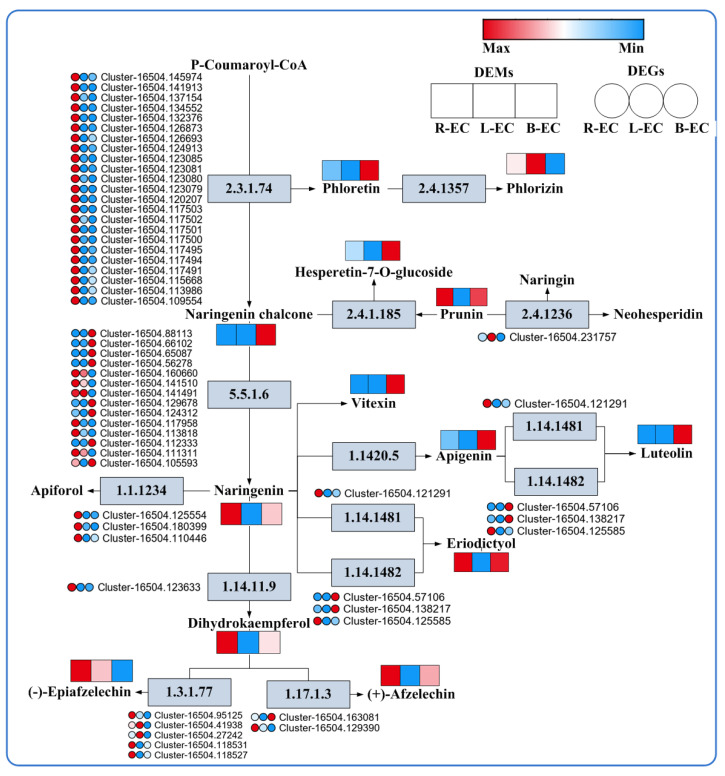
Analysis of DEGs and DEMs of flavonoid metabolic pathways among Korean pine R-EC, L-EC, and B-EC cell lines. 2.3.1.74: chalcone synthase; 2.4.1236: flavanone 7-O-glucoside 2″-O-β-L-rhamnosyltransferase; 5.5.1.6: chalcone iso 1.1.1234: flavanone 4-reductase; 1.14.1481: flavonoid 3′, 5′-hydroxylase; 1.14.1482: flavonoid 3′-monooxygenase; 1.14.11.9: pomelo peel 3-dioxygenase; 1.3.1.77: anthocyanin reductase; 1.17.1.3: leucocyanidin reductase.

**Table 1 ijms-23-13301-t001:** Sequencing data evaluation and statistical results of different somatic embryogenesis potential cell lines of Korean pine.

Sample	Raw Reads	Clean Reads	Clean Bases (G)	Error Rate (%)	Q20 (%)	GC Content (%)
R-EC1	57,338,114	55,752,360	8.36	0.03	97.12	44.34
R-EC2	55,311,330	54,021,080	8.1	0.03	97.07	44.19
R-EC3	61,795,852	60,159,682	9.02	0.03	97.18	44.17
L-EC1	60,854,538	59,090,214	8.86	0.03	97.19	44.27
L-EC2	70,881,322	68,699,534	10.3	0.03	97.19	44.16
L-EC3	56,300,036	54,869,884	8.23	0.03	97.06	44.22
B-EC1	65,271,562	63,299,676	9.49	0.03	97.11	44.28
B-EC2	61,163,512	59,224,850	8.88	0.03	97.17	44.27
B-EC3	73,620,870	70,887,378	10.63	0.03	97.14	44.27

Note: Raw Reads: the number of reads in the original data; Clean Reads: the number of high-quality reads after filtering the original data; Clean Bases: the total number of bases of the high-quality reads; Error Rate: the overall sequencing error rate; Q20: Qphred value; a number of bases no less than 20 account for the percentage of total bases; GC Countent: the sum of the number of G and C in high-quality reads accounts for the percentage of total bases.

**Table 2 ijms-23-13301-t002:** Statistics of splicing results of Korean pine transcripts.

Type	Number	Mean Length	N50	N90	Total Bases
Transcript	325,415	959	1718	366	312,096,367
Unigene	264,676	1115	1842	459	295,135,923

Note: Mean length: the average length of the sequence; N50/N90: sort the spliced transcripts in order of length from long to segment, and add the length of the transcript to the length of the spliced transcript that is no less than 50% and 90% of the total length; Total Bases: the total number of bases in the sequence.

**Table 3 ijms-23-13301-t003:** Differences in gene expression of the WRKY, MYB, and AP2/ERF-ERF families of genes among different Korean pine samples.

Sample	WRKY Family		MYB Family		AP2/ERF-ERF Family	
	up	down	up	down	up	down
R-EC_vs._L-EC	0	4	1	13	2	14
R-EC_vs._B-EC	4	15	6	15	27	26
B-EC_vs._L-EC	6	13	13	27	20	26

## Data Availability

All RNA-seq reads were deposited at NCBI (BioProject ID: SUB12002376).

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
