# Peer review of "Transcriptomic and Metabolomic Analysis of Korean Pine Cell Lines with Different Somatic Embryogenic Potential"

_ijms, 2022, doi:10.3390/ijms232113301_

Round 1

Reviewer 1 Report

Study entitled "Transcriptome and metabolome analysis among different somatic embryogenic potential cell lines in Korean pine" sounds interesting. 

But I have some major and minor concerns with this study.

Minor concerns

1. In introduction section details about the importance and significance of Korean pine is missing. Author focussed on on technique only. Provide details of origin and importance of selected organism along with interest of study.

2. In Material and method section correct 4.2 as experiment methods.

3. Improve the resolution of figure 2 to understand the applied stat. Also in figure 2, what is risk factor? explain briefly.

Major concerns

1. Why author choose sample material only at 7 days of embryonic tissue proliferation for all the tissues?. Is there any control for comparative analysis?

2. Author used de-novo assembly but details of assembly such as N50 values, number of assembled sequences with average length is missing. 

3. How author annotated the sequences and identified transcription factors?

4. What proportion of assembled sequences is still uncharacterised?

5. Why author is not comparing different stages of each cell line to avoid false discovery rates of expressed genes?

Author Response

Dear Reviewer,

Our sincere thanks to you for the time and effort that you have put into reviewing our manuscript! We found all the comments very constructive and helpful, and have revised our manuscript according to all comments. Please find, below, our point-by-point response to the comments raised.

Thank you for considering our revised manuscript!

Point 1: In introduction section details about the importance and significance of Korean pine is missing. Author focussed on on technique only. Provide details of origin and importance of selected organism along with interest of study.

Response 1: We have added to the importance of some Korean pines. Please see the lines 40 and 41.

Point 2: In Material and method section correct 4.2 as experiment methods.

Response 2: We have made changes. Please see the line 407.

Point 3: Improve the resolution of figure 2 to understand the applied stat. Also in figure 2, what is risk factor? explain briefly.

Response 3: We have modified Figure 2, which shows the rich factor. Please see the line 108 (It is the Figure3).

Point 4: Why author choose sample material only at 7 days of embryonic tissue proliferation for all the tissues? Is there any control for comparative analysis?

Response 4: We selected a 7-day-old embryonic tissue sample as the experimental material, because it will be used for somatic embryo maturation. There were three cell lines were used as research materials in this article. They were all induced by the same 1# family. They were cultivated by the same way for the same days. We compared these cell lines. Please see the lines 396-401.

Point 5: Author used de-novo assembly but details of assembly such as N50 values, number of assembled sequences with average length is missing.

Response 5: In the article, we added the N50 value, which is the average length of the assembly sequence number. Please see the Table 2 (lines 99-102).

Point 6: How author annotated the sequences and identified transcription factors?

Response 6: The annotated sequences and identification of transcription factors are done by specialized biological companies. Please see the table and figure as follow.

Point 7: What proportion of assembled sequences is still uncharacterised?

Response 7: We supplemented some important results in our article. Please see the lines 84-104.

The transcriptomes of R-EC, B-EC and L-EC were analyzed to investigate the molecular events between different embryogenic calli. After removing low-quality readings, a total of 546004658 clean readings were obtained. The detection rates of Q20 and GC were 97.06-97.19% and 44.16-44.27%, respectively, indicating higher quality of transcriptome sequencing data (Table 1).

Table 1 Sequencing data evaluation and statistical results of different somatic embryogenesis potential cell lines of Korean pine

Sample

Raw Reads

Clean Reads

Clean Barse

 (G)

Error Rate

 (%)

Q20

(%)

GC Content

(%)

R-EC1

57338114

55752360

8.36

0.03

97.12

44.34

R-EC2

55311330

54021080

8.1

0.03

97.07

44.19

R-EC3

61795852

60159682

9.02

0.03

97.18

44.17

L-EC1

60854538

59090214

8.86

0.03

97.19

44.27

L-EC2

70881322

68699534

10.3

0.03

97.19

44.16

L-EC3

56300036

54869884

8.23

0.03

97.06

44.22

B-EC1

65271562

63299676

9.49

0.03

97.11

44.28

B-EC2

61163512

59224850

8.88

0.03

97.17

44.27

B-EC3

73620870

70887378

10.63

0.03

97.14

44.27

Note: Raw Reads: the number of reads in the original data, Clean Reads: the number of high-quality reads after filtering the original data, Clean Bases: the total number of bases of the high-quality reads, Error Rate: the overall Sequencing error rate, Q20: Qphred value The number of bases not less than 20 accounts for the percentage of total bases, GC Countent: The sum of the number of G and C in high-quality reads accounts for the percentage of total bases.

A total of 264676 Unigenes were assembled from clean sequences using Trinity, of which 1842 were N50 non-redundant sequences and 459 were N90 non-redundant sequences (Table 2), the 300-400 bp length size accounted for the largest proportion, and simultaneously assembled transcript and Unigene decreased gradually with increasing sequence length (Fig.2)

Table 2 Statistics of splicing results of Korean pine transcripts

Type

Number

Mean Length

N50

N90

Total Bases

Transcript

325415

959

1718

366

312096367

Unigene

264676

1115

1842

459

295135923

Note: Mean length: the average length of the sequence, N50/N90: Sort the spliced transcripts in order of length from long to segment, and add the length of the transcript to the length of the spliced transcript that is not less than 50% and 90% of the total length, Total Bases: The total number of bases in the sequence.

Figure 2 Splicing sequence length distribution map of Korean pine transcript

Point 8: Why author is not comparing different stages of each cell line to avoid false discovery rates of expressed genes?

Response 8: We supplemented some important informations in this article. Please see the lines 396-401. The plants used in this study complies with international guidelines. Full sibling family cones 1# were authorized to be collected from cooperative institution (Korean pine seed orchard of Lushuihe Forestry Bureau of Jilin Province) on July 1, 2018. The SE cell line (R-EC), blocked SE cell line (B-EC), and the loss of SE cell line (L-EC) were used as research materials. They were all induced by 1# family. We selected a 7-day-old embryonic tissue sample as the experimental material, because the state of the embryogenic callus at the proliferative stage already determines the outcome of subsequent somatic embryogenesis, many results can be obtained by comparing the embryogenic callus at the proliferative stage.

Reviewer 2 Report

There is abundance of misspellings throughout the article. The English, the writing, and the tenses should be improved in certain sections, such as the methodology. The manuscript requires a thorough revision of the English. As an example, this reviewer has edited the abstract.

Abstract

The embryogenesis capacity of conifer callus is not only highly genotype-dependent but also gradually lost after long-term proliferation. These problems have seriously limited the commercialization of conifer somatic embryogenesis (SE) technology commercialization. In this study, responsive SE cell line (R-EC), blocked SE cell line (B-EC), and the loss of SE cell line (L-EC) were selected as the research objects. The morphological, physiological, transcriptome, and metabolome analysis of three types of cells was were performed. The results showed that R-EC had higher water content, total sugar content, and putrescine (Put) content, and ,lower SOD activity and H2O2 content than B-EC and L- EC. 2566, 13768, and 13900 differentially expressed genes (DEGs) and 219, 253, and 341 differentially expressed metabolites (DEMs) were found between R-EC vs. B-EC, R-EC vs. B-EC and B-EC vs. L- EC, respectively. These DEGs and DEMs are mainly involved in plant signal transduction, starch and sugar metabolism, phenylpropane metabolism and flavonoid metabolism pathways and flavonoid metabolism. We found that AUX1 and AUX/IAA family genes were significantly up-regulated after the long-term proliferation of callus, and the auxin content was higher[VMLV1] . Most of the phenylpropane and flavonoid metabolites, which act as antioxidants to protect themselves from damage, were found to be significantly up-regulated in R-EC.

Comments

Scientific names should be in italics.

In some parts, the authors talk about the embryogenic potential of their lines, and in others, they talk about somatic embryogenesis. It is not very clear, and it seems that the authors are not clear about where the process of somatic embryogenesis begins.

This reviewer has many doubts about handling the data because sometimes they talk about differential expression and they are not comparing the lines. Instead, it would seem that they compare the expression (without being differential) of each line by itself. If it is differential, it is because there is a point of comparison.

Everything is very descriptive. Perhaps in this work, it would be essential to capture their results in a short and straightforward model where they concentrate only on the most relevant that distinguishes one line from another. In the abstract, they mention the Aux/IAA genes, and in the conclusion, it went unnoticed. Between this and other results, the conclusion of his work is unclear to this reviewer.

Some specific comments are listed below, and the line number in which they are found.

In Figure 1B, the third graph does not have the statistic letters on the POD activity bars.

Line 92: correct GC "content."

Line 95: "to find overlapping genes during somatic embryogenesis," but they did not do somatic embryogenesis.

Line 98: not six different tissues were analyzed, but three and the combinations were generated to compare them.

Figure 1: They do not mention the statistical tests they did or the value of P.

Figure 2B and 6A and 6B: verify that the way of writing the comparisons is homogeneous (put "vs.", not just "v").

Figure 3: The third column is poorly colored, and the upregulated are missing. Figure 4: Not understood. I guess it is similar to a heatmap. It is missing to put the meaning of the color scale.

Figure 6: The footer says "statistical analysis," but you did not see any statistics other than the count. Metabolites are not expressed. Differentially expressed metabolites could be changed to "differentially accumulated metabolites." This implies changing it throughout the entire article.

Figure 7: At the top in the left corner, it says "Fig. 6" when it is 7. Perhaps it would be worth putting the name of each of the genes if the annotation was made. Correct "cytokinin."

Figure 8: You also have the wrong figure number in the upper left corner.

Discussion:

They mention that they tested during the somatic embryogenesis process but did so on embryogenic callus lines (line 282 and most subsequent sections of the discussion).

Lines 359-360: There are many sentences without connection.

Lines 368-371: On the other hand, this reviewer thinks that in addition to the fact that the discussion in this paragraph is not well focused, it is not very reliable to make a relationship with their study because they did not directly evaluate lignin. In any case, the importance of lignin in ES (or in embryogenic calli) goes beyond resistance to pathogens. There are already many articles reporting results in this regard that would be more useful for the discussion.

Line 377: "differentially metabolized genes"?

Lines 386-387: Which studies? They do not quote them.

Lines 388-393: They repeat the same thing they had already said in the results. There is no discussion about these findings or support for their proposed function.

Materials and methods:

Section 4.1 does not mention the study model (although it appears in the article's title). In line 398, the sentence could be rephrased to be understood. In general, they need to describe the culture conditions, how the lines were generated if they did it, or cite an article where they have done it.

Line 404: Must be "Experimental method"

Line 416: correct "residue"

Lines 432 and 433: the sentence is in the present tense.

Line 443: correct RNA-seq (with a hyphen).

Line 444: correct the first link because it does not open.

Lines 446-448: Rephrasing because it is not understood. It is all together. You also need to specify the parameters used for the differential expression (the fold change and FDR) and the functional annotation. Everything is very general.

Line 451: The article they cite for the methodology of the metabolic analysis also does not explain how it was done, apart from the fact that the paragraph is copied the same. This reviewer thinks they got the article wrong because it is by the same author and the same year. They should cite this article: Yuan et al. 2018, DOI: 10.1155/2018/9415409.

It is missing to place the link to where the transcriptome and metabolome data were uploaded to some repository.

 Incomplete comparison. It appears that this sentence includes an incomplete comparison. Consider rewriting to complete the comparison.

Author Response

Dear Reviewer,

Our sincere thanks to you for the time and effort that you have put into reviewing our manuscript! We found all the comments very constructive and helpful, and have revised our manuscript according to all comments. Please find, below, our point-by-point response to the comments raised.

Thank you for considering our revised manuscript!

Point 1: There is abundance of misspellings throughout the article. The English, the writing, and the tenses should be improved in certain sections, such as the methodology. The manuscript requires a thorough revision of the English. As an example, this reviewer has edited the abstract.

Response 1: We completely revised the English of the whole article. The authors would like to thank TopEdit (www.topeditsci.com) for its linguistic assistance during the preparation of this manuscript.

Point 2: Scientific names should be in italics.

Response 2: Corrections have been made for scientific names.

Point 3: In some parts, the authors talk about the embryogenic potential of their lines, and in others, they talk about somatic embryogenesis. It is not very clear, and it seems that the authors are not clear about where the process of somatic embryogenesis begins.

Response 3: In our study, we focused on the somatic embryogenic potential of different types of embryogenic calli. Embryogenic Callus is an important part of somatic embryogenesis, so somatic embryogenesis is also discussed.

Point 4: In Figure 1B, the third graph does not have the statistic letters on the POD activity bars.

Response 4: Done. Please see above.

Point 5: Line 92: correct GC "content."

Response 5: Table 1 has been deleted.

Point 6: Line 95: "to find overlapping genes during somatic embryogenesis," but they did not do somatic embryogenesis.

Response 6: “We created Venn diagrams to find overlapping genes during somatic embryogenesis in 3 different types of Korean pine callus” has been replaced by “We created Venn diagrams to find some overlapping genes in 3 different types of Korean pine callus”.

Point 7: Line 98: not six different tissues were analyzed, but three and the combinations were generated to compare them.

Response 7: “six” has been replaced by “three”.

Point 8: Figure 1: They do not mention the statistical tests they did or the value of P.

Response 8: Done. Please see above.

Point 9: Figure 2B and 6A and 6B: verify that the way of writing the comparisons is homogeneous (put "vs.", not just "v").

Response 9: Done. Please see above.

Point 10: Figure 3: The third column is poorly colored, and the upregulated are missing. Figure 4: Not understood. I guess it is similar to a heatmap. It is missing to put the meaning of the color scale.

Response 10: We changed the color of Figure 3. Figure 4 adds a color scale.

Point 11: Figure 6: The footer says "statistical analysis," but you did not see any statistics other than the count. Metabolites are not expressed. Differentially expressed metabolites could be changed to "differentially accumulated metabolites." This implies changing it throughout the entire article.

Response 11: Done. Please see above.

Point 12: Figure 7: At the top in the left corner, it says "Fig. 6" when it is 7. Perhaps it would be worth putting the name of each of the genes if the annotation was made. Correct "cytokinin."

Response 12: Done. Please see above.

Point 13: Figure 8: You also have the wrong figure number in the upper left corner.

Response 13: Done. Please see above.

Point 14: Discussion:They mention that they tested during the somatic embryogenesis process but did so on embryogenic callus lines (line 282 and most subsequent sections of the discussion).

Response 14: In our experiments, R-EC is a cell line that can normally obtain normal somatic embryos. Embryogenic callus is an important stage in the process of somatic embryogenesis, so in the Discussion we also compared the results of some other species tested during somatic embryogenesis.

Point 15: Lines 359-360: There are many sentences without connection.

Response 15: We have corrected the sentence.

Point 16: Lines 368-371: On the other hand, this reviewer thinks that in addition to the fact that the discussion in this paragraph is not well focused, it is not very reliable to make a relationship with their study because they did not directly evaluate lignin. In any case, the importance of lignin in ES (or in embryogenic calli) goes beyond resistance to pathogens. There are already many articles reporting results in this regard that would be more useful for the discussion.

Response 16: We discuss the role of lignin in plant secondary cell walls.

Point 17: Line 377: "differentially metabolized genes"?

Response 17: “differentially metabolized genes” has been replaced by “differentially expressed genes”.

Point 18: Lines 386-387: Which studies? They do not quote them.

Response 18: We have added references.

Point 20: Lines 388-393: They repeat the same thing they had already said in the results. There is no discussion about these findings or support for their proposed function.

Response 20: For this part, we have revisited the discussion.

Point 21: Section 4.1 does not mention the study model (although it appears in the article's title). In line 398, the sentence could be rephrased to be understood. In general, they need to describe the culture conditions, how the lines were generated if they did it, or cite an article where they have done it.

Response 21: Section 4.1 has been revised and references added.

Point 22: Line 404: Must be "Experimental method"

Response 22: Done. Please see above.

Point 23: Line 416: correct "residue"

Response 23: Done. Please see above.

Point 24: Lines 432 and 433: the sentence is in the present tense.

Response 24: Done. Please see above.

Point 25: Line 443: correct RNA-seq (with a hyphen).

Response 25: Done. Please see above.

Point 26: Line 444: correct the first link because it does not open.

Response 26: Done. Please see above.

Point 27: Lines 446-448: Rephrasing because it is not understood. It is all together. You also need to specify the parameters used for the differential expression (the fold change and FDR) and the functional annotation. Everything is very general.

Response 27: We rewrite the conclusion.

Point 28: Line 451: The article they cite for the methodology of the metabolic analysis also does not explain how it was done, apart from the fact that the paragraph is copied the same. This reviewer thinks they got the article wrong because it is by the same author and the same year. They should cite this article: Yuan et al. 2018, DOI: 10.1155/2018/9415409.

Response 28: We have corrected the references.

Point 29: It is missing to place the link to where the transcriptome and metabolome data were uploaded to some repository.

Response 29: We have added it in the article.

Reviewer 3 Report

Comments to the Authors

The review on the paper by Chunxue Peng et al., “Transcriptome and metabolome analysis among different somatic embryogenic potential cell lines in Korean pine”

The aim of the study is to characterize the physiological and molecular events related to somatic embryo productivity and to explore the key factors that determine the genotype-dependent somatic embryogenesis ability and its decreasing after long-term proliferation in Korean pine. Authors selected three genotypically different cell lines differed by their embryogenesis capacity and after 7 days of proliferation run the morphological, physiological, transcriptome and metabolome analysis of the collected tissues. Authors suggested that any differences they will found between samples would explain the distinctions in the embryogenic abilities of calluses.

In general, it is methodologically simple, descriptive paper, defining the set of morphological, physiological, transcriptomic and metabolomic characters differing cell lines. Authors made a great work to characterize selected cell lines with multiple morphological and molecular traits. But the defined differences in unknown degree could be stipulated by genetic differences and not the embryogenic ability. The obtained results are not very clear and could be not the reason but the results on different embryogenic abilities of the genotypes. Considerably more elegant experiment could be realized with the genetically identical responsive cell lines, then run several rounds of proliferation until they will reach the blocked state (if they …), looking for the intermediate loss-states within the same genotypes.

I would suggest below some general and specific comments that may hopefully help the authors to reshape their manuscript in a form more accessible to the readers and revealing its scientific essentials.

I did not find any info about transcriptomic data uploaded to the public resources.

All the transcriptomic analyses were done completely in silico, it would be wise to validate some of them using at least qRT-PCR. No information presented about transcripts assembly and gene models definition. As Korean pine genome is not sequenced yet, these data could be very useful for the scientific community

R-EC, B-EC and L-EC abbreviations look very heavy, especially when all three presented together, and making difficulties in reading the text. They could be replaced by word descriptions, e.g. all three cell lines …, studied cell lines, etc.

All the explanations of abbreviations should be done in main text and not in the abstracts.

Table 1 should be moved into supplements or removed; it is not related to the topic of paper. Information about defined gene models could be more relevant

Fig.2 Venn diagram looks very strange. I did not understand what means unique genes between R-EC vs B-EC, etc. It could have sense for DEGs, but not for the transcripts description or it should be explained more clear.

ll. 133& ll. 296, etc - according to reports … statements and similar, should be followed by corresponding references

ll. 127 – it would be nice to present here the general information about defined TFs, total number, different classes, etc. Not it looks that different cell lines have different number of transcription factors.

Fig. 4. What is the difference between 4c and 4d?

In the discussion part, obtained results in most cases were referred to crop and angiosperm species. There is the huge amount of publication for somatic embryogenesis in coniferous species. It would be more relevant to use them also for discussion for Korean pine.

ll 430 … samples of the samples … should be somehow corrected.

In conclusion it looks that authors overestimate the obtained results. I would recommend avoiding superior terms or make them more concrete, which findings affect theoretical understandings of in vitro propagation.

Author Response

Dear Reviewer,

Our sincere thanks to you for the time and effort that you have put into reviewing our manuscript! We found all the comments very constructive and helpful, and have revised our manuscript according to all comments. Please find, below, our point-by-point response to the comments raised.

Thank you for considering our revised manuscript!

Point 1: All the transcriptomic analyses were done completely in silico, it would be wise to validate some of them using at least qRT-PCR. No information presented about transcripts assembly and gene models definition. As Korean pine genome is not sequenced yet, these data could be very useful for the scientific community

Response 1: This article was not validated using qRT-PCR.

Point 2: R-EC, B-EC and L-EC abbreviations look very heavy, especially when all three presented together, and making difficulties in reading the text. They could be replaced by word descriptions, e.g. all three cell lines …, studied cell lines, etc.

Response 2: When these three abbreviations are put together, we have made changes.

Point 3: All the explanations of abbreviations should be done in main text and not in the abstracts.

Response 3: We have corrected the interpretation of abbreviations.

Point 4: Table 1 should be moved into supplements or removed; it is not related to the topic of paper. Information about defined gene models could be more relevant

Response 4: we have deleted table 1.

Point 5: Fig.2 Venn diagram looks very strange. I did not understand what means unique genes between R-EC vs B-EC, etc. It could have sense for DEGs, but not for the transcripts description or it should be explained more clear.

Response 5: We made corrections.

Point 6: ll. 133& ll. 296, etc - according to reports … statements and similar, should be followed by corresponding references

Response 6: We have added references.

Point 7: ll. 127 – it would be nice to present here the general information about defined TFs, total number, different classes, etc. Not it looks that different cell lines have different number of transcription factors.

Response 7: Because transcription factors are not the focus of this article, no general information is provided about the defined TFS, total number, different classes, etc.

Point 8: Fig. 4. What is the difference between 4c and 4d?

Response 8: For a neat layout, we split a figure into two figures, figure-4c and figure-4d.

Point 9: In the discussion part, obtained results in most cases were referred to crop and angiosperm species. There is the huge amount of publication for somatic embryogenesis in coniferous species. It would be more relevant to use them also for discussion for Korean pine.

Response 9: There are relatively few studies on the embryogenic potential of conifers somatic cells, and the literature is not much cited. We supplement some literature on conifer somatic embryogenesis.

Point 10: ll 430 … samples of the samples … should be somehow corrected.

Response 10: Done. Please see above.

Point 11: In conclusion it looks that authors overestimate the obtained results. I would recommend avoiding superior terms or make them more concrete, which findings affect theoretical understandings of in vitro propagation.

Response 11: We have rewritten our conclusions.

Round 2

Reviewer 1 Report

I recommend to accept this manuscript. Author need to provide high resolution figures so that reviewers can understand in better way.

Reviewer 2 Report

The authors addressed all the changes required